# Evaluation of Effect of Thermoplastic Polyurethane (TPU) on Crumb Rubber Modified (CRM) Asphalt Binder

**DOI:** 10.3390/ma15113824

**Published:** 2022-05-27

**Authors:** Jihyeon Yun, Mithil Mazumder, Il-Ho Na, Moon-Sup Lee, Hyun Hwan Kim

**Affiliations:** 1Department of Engineering Technology, Texas State University, San Marcos, TX 78666, USA; yiy1@txstate.edu (J.Y.); m_m624@txstate.edu (M.M.); 2Korea Petroleum, Seoul 04427, Korea; ihna@koreapetroleum.com; 3Korea Institute of Civil Engineering and Building Technology, Goyang 10223, Korea

**Keywords:** crumb rubber, thermoplastic polyurethane, viscosity, rutting, cracking

## Abstract

Crumb rubber binder with thermoplastic polyurethane (TPU) has been experimented with to characterize the performance properties considering the workability, rutting, fatigue cracking and cracking resistance at low temperatures depending on the temperatures and aging states. Physical and rheological properties were evaluated to proceed with the study by applying Superpave asphalt binder testing and multi-stress creep recovery (MSCR). Based on the targeted experiments, the binder samples were produced at three aging states (original, short-term aged and long-term aged) using a rolling thin film oven (RTFO) and pressure aging vessel (PAV). The results revealed that (i) the addition of TPU into CRM binders has a potential effect on increasing viscoelasticity at the original condition, (ii) CRM binders containing TPU showed improved anti-aging performance based on results of RTFO residues and (iii) the inclusion of TPU made it possible for CRM asphalt binder to improve its fatigue and cracking resistance at low temperature.

## 1. Introduction

Due to the rapid development of industry around the world, increasing traffic volume and extreme climate change are becoming the main problem for the service life of asphalt pavement [1,2]. This has led to considering high-performance asphalt binders as a solution to contribute to service life [3,4]. However, even though using modifiers is beneficial to asphalt materials, the aging of asphalt binders affects almost all aspects of asphalt pavement materials, making aging a fundamentally important factor that affects the performance and durability of asphalt pavement [5]. Therefore, in order to make high-performance asphalt mixtures, it is necessary for the modified binder to be considered with aging resistance [6].

Aging of asphalt binder proceeds in two stages: short-term aging in the process of mixing, transporting and paving, and long-term aging, which occurs during service life in the field. Short-term aging is mainly related to the loss of volatile substances and rapid oxidation at high temperatures, whereas long-term aging is more associated with gradual oxidation, steric hardening and ultraviolet radiation [7]. In the case of short-term aging of asphalt binders, if an asphalt mixture with severe short-term aging is paved, there is a high possibility that the service life will be shortened due to a sudden increase in brittleness [8]. This means the asphalt pavement will remain in a relatively high aging state, which will be accelerated with time. Therefore, the service life will be shortened. Such problems encountered during the work will give significant financial and economic losses. 

Crumb rubber modifier (CRM) is one of the popular asphalt modifiers which has been widely introduced and utilized to enhance asphalt binder performance. In addition, the other benefit of using this waste material is a solution to the environment and landfill problems. This is applied in the pavement industry as a valuable recycling modifier. CRM binder can improve the rutting resistance as it increases the properties of stiffness and elasticity at a high temperature [9]. Moreover, it provides an extended life cycle of the asphalt mixture. On the other hand, thermoplastic polyurethane (TPU) is one of the modifiers that are insufficiently studied and there is not enough inspection and evaluation carried out concerning its applications in the asphalt industry [10,11]. TPU may be used to replace rubber, plastic and metal due to its light weight, low noise, corrosion resistance and high economic value, suggesting that TPU can be applied in many industries [12,13]. Moreover, TPU as a polymer modifier can improve the flexibility and strength of modified asphalt binder [14,15]. In addition, its compatible property can be improved by the reaction between the functional group of TPU and the basic element of asphalt, improving the rheological properties of rutting and cracking performance [1,16,17,18]. In recent years, researchers have examined the effects of TPU on SIS asphalt binder. In this study, they found that the TPU was effective in improving the resistance to cracks occurring in the brittleness of the binder at low temperatures [19]. In summary, the purpose of this study is to assess the possibility of improving the characteristics of asphalt binder containing CRM and TPU. In order to evaluate the asphalt binder performance, the Superpave binder test and MSCR were conducted. Figure 1 shows a flow chart of the experimental design used in this study

## 2. Material and Test Program

### 2.1. Materials

A performance grade (PG) 64-22 asphalt binder was adopted as the base binder and utilized to modify ground tire rubber and thermoplastic polyurethane. Table 1 and Table 2 show the properties of the asphalt binder and gradation of the crumb rubber. The TPU was applied to the CRM binder so as to assess the performance of TPU. The properties of the TPU are shown in Table 3, and Figure 2 shows the CRM and TPU used in this study.

### 2.2. Production of Modified Asphalt Binders

The CRM asphalt binder was produced in the laboratory by adding CRM through a wet process for 30 min with an open blade mixer at 177 °C and a blending speed of 700 rpm [20]. The percentages of crumb rubber used for the rubberized binder were 5%, 10%, 15% and 20% by weight of the base binder. In order to use and evaluate the TPU, this material was directly added into the CRM asphalt binder and then the TPU was modified for 60 min. To maintain the consistency of the CRM + TPU asphalt binder, only one batch of crumb rubber was applied in this study. 

### 2.3. Superpave Binder Test

In order to measure binders, Superpave binder tests were applied to assess the performance. Before going into each test, modified binders were conditioned in three stages: original, short-term aging (Rolling Thin film Oven procedure, RTF 325-B, James cox & sons Inc., Colfax, CA, USA) and long-term aging (Pressure Aging Vessel) conditions. Rotational viscosity (Test Method D4402), dynamic shear rheometer (Test Method D7175), multiple stress creep recovery (Test Method D7405) and bending beam rheometer (AASHTOT 313) were conducted. Figure 3 reflects the procedure of the Superpave binder test conducted in this study.

The rotational viscosity test was conducted at 135 °C by selecting 27 cylindrical spindles and a constant speed of 20 rpm with a weight of 10.5 g of the original binder sample. The time to acquire data was considered to be 20 min for each sample. To measure the viscoelasticity of the asphalt binder, G*/sin *δ* was calculated from the complex shear modulus (G*) and the sine (*δ*) of the phase angle at 70 °C, using binders of original and short-term aged samples. For RTFO + PAV binders, the fatigue cracking property and the cracking property at low temperatures were evaluated by G*sin *δ* and stiffness results. G*sin *δ* was measured at the intermediate temperature of 25 °C to evaluate fatigue cracking properties. In order to assess cracking properties at low temperature, asphalt beams (125 × 6.35 × 12.7 mm) were formed to perform the BBR test. The beam was supported at both ends, and the stiffness was calculated by applying a constant load of 100 g to the center of the beam at a temperature of −12 °C.

### 2.4. Multiple Stress Creep Recovery (MSCR)

DSR was operated to conduct the multi-stress creep and recovery (MSCR) test for the asphalt binder. The MSCR test was performed at 64 °C, loading 3.2 kPa onto the RTFO binder sample in order to evaluate viscoelasticity properties. By investigating the MSCR test, two parameters of the nonrecoverable creep compliance (J_nr_) and percent recovery (%rec) were derived. In particular, the nonrecoverable creep compliance (J_nr_), which is the result of the non-recoverable shear strain divided by the shear stress, is used to assess the viscoelasticity of the asphalt binder. Figure 4 shows the binder is evaluated for creep loading and unloading cycles of 1 s and 9 s at stress levels of 3.2 kPa loading ten cycles.

### 2.5. Statistical Analysis

Statistical analysis was conducted adopting the Statistical Package for the Social Sciences (SPSS) program to conduct an analysis of variance (ANOVA), applying Fisher’s Least Significant Difference (LSD) comparison with α = 0.05. ANOVA was first performed to demonstrate whether there was a significant difference between means for samples. In this analysis, the significance level is 95, showing a 95% chance that each outcome is true. After running ANOVA, LSD is calculated and defined as the observed difference between two sample means required to confirm the difference between the corresponding population means. All pairs of sample means were statistically classified if the difference between the means was greater than or equal to the LSD [21].

## 3. Results and Discussions

### 3.1. Rotational Viscosity

The viscosity of the asphalt binder plays an important role in constructing a new asphalt pavement as it affects the workability from production and transportation to compaction. Figure 5 shows the viscous value at 135 °C for the asphalt binder containing CRM and TPU. The viscosity increased gradually with the addition of CRM, with an increasing value from approximately 500 cP to 2400 cP. In addition, all viscosity shown in the bar chart increased with the addition of 5% TPU. It is observed that increasing TPU contents contributed to raising the viscosity values. Even though the addition of TPU is effective in increasing the binder viscosity, it is much less compared to CRM. It might be beneficial to apply TPU in terms of workability if TPU shows positive effects on other parameters of binder performance, given its low impact on increasing the viscosity of the binder. All results indicated lower values than the maximum specification (3000 cP) at 135 °C suggested by Superpave. 

The statistical significance of the CRM binder by using a one-way analysis of variance was evaluated depending on whether the TPU was added or not (Table 4). Overall, there was a statistically significant difference among all viscosity values. In particular, it was evident that the viscosity values showed a significant difference according to increasing CRM particles up to 15%. In addition, the statistical significance of using TPU was verified in the CRM binders.

### 3.2. Dynamic Shear Rheometer Test

#### 3.2.1. Original G*/Sin *δ*

DSR (Test method D 7175) is one of the most common tests used for the rutting property. G*/sin *δ* is a value indicating the rutting property for the binder, and the higher the G*/sin *δ*, the higher the rutting resistance. The DSR results are shown in Figure 6. Basically, the G*/sin *δ* value at 70 °C increased due to the increasing content of the CRM modifier, which means CRM is effective at developing rutting resistance. In addition to that, the inclusion of TPU was also observed to increase the G*/sin *δ* value. This indicates that the addition of TPU is also advantageous in terms of improving the rutting resistance of CRM binders. 

The statistical significance of the CRM binders using the TPU or not with an increase in testing temperature was examined using a one-way analysis of variance. The results are shown in Table 5. In general, a significant difference among all binders was observed at all temperatures. In addition, as with the statistical analysis for the rotational viscosity, a significant difference was shown using the TPU in all CRM binders. This indicates that the TPU plays a role in increasing G*/sin *δ*.

#### 3.2.2. RTFO G*/Sin *δ*

After conditioning binders with the RTFO procedure, short-term aged binders were sampled and evaluated in the same way as testing the original G*/sin *δ* at 70 °C. All results of G*/sin *δ* presented an increasing trend with the addition of CRM, which is recognized to be similar to that of the original G*/sin *δ* as shown in Figure 7. However, it is worth noting that, unlike the trend for original G*/sin *δ*, when TPU is added to the CRM binder, the G*/sin *δ* in RTFO showed a lower value compared to the corresponding control CRM binder. In general, the asphalt binder has a hard portion increased by the aging process. Because of the increased hard portion, the aged binders show higher G*/sin *δ* and are considered to have higher rutting resistance. Superpave suggests a higher minimum value (2.2 kPa) for the RTFO binder, considering this reason. Based on the results, the binder containing TPU has a lesser aging effect than the binders without TPU. Thus, it is expected to positively impact cracking resistance if the TPU is applied to polymer-modified binders. 

Using one-way analysis of variance, the statistical significance of the change in using TPU or not for CRM binders was examined (Table 6). The CRM contents have a significant effect on the G*/sin *δ* values. Although a significant difference was generally seen in all results, some insignificant difference was observed with CRM binders containing TPU. This is because the addition of TPU caused a decrease in the G*/sin *δ* values in short-term aging, with the remaining data concerning use of the TPU being at a similar level to normal CRM data. Therefore, the statistical analysis also showed that the TPU contributes to the anti-aging effect of CRM binders.

#### 3.2.3. Multiple Stress Creep Recovery

MSCR (Test Method D7405) is an alternate test to the DSR test method. The MSCR test was conducted pursuant to AASHTO TP 70 by loading 3.2 kPa to assess the viscoelasticity of the binder under more extreme conditions than the DSR test at 64 °C. 

Generally, MSCR tests are evaluated on RTFO short-term aged binders. However, to analyze the G*/sin *δ* test results more deeply, the original state samples were also used to measure J_nr_ and %rec. The data of the original condition shown in Figure 8 presented the value of J_nr_. In general, the addition of CRM made it possible to decrease the J_nr_ value, which means the more CRM contents, the more viscoelasticity in both aging states. The results of the original state from CRM 0% and 5% binders were not measured at 64 °C, except for the 5% CRM binder containing TPU due to the higher load compared to the G*/sin *δ* test. The J_nr_ value decreased steadily with the addition of TPU into the CRM binder. Based on the results for the original condition, using TPU enhanced the viscoelasticity of CRM binders at the original state. After conditioning the RTFO procedure, as with the results of the original condition, increasing CRM contents contributed to decreasing the J_nr_ value. However, in the case of using the TPU, the J_nr_ value of CRM binders with TPU was relatively higher than general CRM binders, indicating that the addition of TPU into the CRM binder is not effective in improving rutting performances at short-term age. This is similar to G*/sin *δ* results in that RTFO showed lower rutting resistance compared to the original state. 

Figure 9 shows the %rec results of the binders. %rec results showed a trend almost similar to J_nr_; the addition of CRM positively affects the rutting property, and %rec increased with the addition of TPU into the CRM binder. Based on the results for the original condition, the elasticity of CRM binders was enhanced by the application of TPU. In RTFO, all samples were measured to have increased elasticity caused by the short-term aging compared to the original state. On the other hand, in the case of using the TPU, the %rec value of the CRM binder using TPU was relatively lower than general CRM binders in all data. As with the result for the original condition, after short-term aging, the effect of improving rutting resistance of TPU did not appear. As mentioned above, it is considered that TPU is less affected by aging, and thus, the effect of enhancing plastic deformation due to the aging process does not appear. However, it is required that the effect of TPU on cracking performance is evaluated to confirm the aging effect of TPU.

The statistical significance of the change in the J_nr_ and %rec was investigated, comparing the original to the RTFO condition. Overall, the J_nr_ values within the original condition from the MSCR test at 64 °C were significantly different in accordance with the increase of CRM content (Table 7). This indicates that the higher CRM content is more effective in terms of having better rutting performance. In the case of statistical analysis for J_nr_ of CRM binders, including TPU, there was a significant difference compared to general CRM binders. For the RTFO binders, some non-significant difference was observed within using CRM + TPU binders. The reason for the different trends between the original condition and the RTFO condition is that TPU is considered to have a positive effect on anti-aging for short-term aging. The statistical analysis was performed for the %rec result, and it is shown in Table 8. In most cases, it was found that there was a significant difference in the content of CRM and the addition of TPU. In particular, it was statistically significant that the 10% and 15% CRM binders with 5% TPU showed lower %rec than CRM binders without TPU in short-term aging. This means that the aging effect on TPU, which was mentioned above, is statistically significant.

#### 3.2.4. PAV G*Sin *δ*

In general, the lower G*sin *δ* values are recognized as a benefit concerning fatigue cracking resistance. The G*sin *δ* values of binders containing CRM and TPU were evaluated by using DSR at 25 °C. Figure 10 depicts the results of G*sin *δ* in this study. The tendency for gradually decreasing G*sin *δ* values was witnessed from approximately 4500 kPa to 2500 kPa, with the addition of CRM contents. It is obvious that the application of CRM is effective at decreasing the G*sin *δ* value. In addition, the CRM binders with TPU revealed a lower G*sin *δ* value compared to their corresponding control CRM binder. Based on the results, both additives positively influenced the fatigue cracking performances of asphalt binders. In the above-described permanent deformation resistance evaluation test (G*/sin *δ* and MSCR test), TPU, which was shown to improve the performance of the binder before aging, was found to have lower rutting resistance to the binder after short-term aging than the control CRM binders without TPU. However, in tests performed with samples subjected to long-term aging, all TPU specimens indicated higher cracking resistance than samples without TPU. In general, concerning the relation to the aging of the binder, the hard portion in the binder increases after the aging process. It shows high resistance to deformation, whereas, in the case of TPU, the increase in the hard portion in the binder is reduced even if aging is experienced. It is considered that the effect has a positive influence on cracking that behaves differently from plastic deformation. The statistical significance for the G*sin *δ* value depending on the addition of CRM and TPU was investigated. The results are shown in Table 9. The results revealed that CRM plays a role in the significant effect on G*sin *δ* value. Furthermore, binders containing the CRM and TPU were discovered to have a statistically significant difference for the G*sin *δ* value by the 5% confidence level. There was no statistically significant difference within binders of CRM10%, CRM 0% and CRM5% + TPU 5%, which means, however, that the TPU effect decreases, which in turn decreases the G*sin *δ* value, the two remaining relatively similar to one other.

### 3.3. Bending Beam Rheometer (BBR) Test

From Superpave requirements, the maximum value of 300 MPa for stiffness is proposed; the results come from using the DSR test. The decreased stiffness of the asphalt binder is predicted to promote smaller stress and cracking at low temperatures. The stiffness of CRM binders containing TPU was evaluated at −12 °C by the BBR tests, and Figure 11 illustrates the results. The results revealed that using CRM made it possible to steadily decrease the stiffness value, with a low of 122.5 MPa in CRM 15% from 188 MPa in CRM 0%. In particular, the stiffness for the binder containing TPU remained relatively lower than normal CRM binders in each content. This trend is the same with the fatigue cracking property results. On the basis of the results of the data, it is considered that TPU can play a role in raising the thermal cracking resistance at low temperatures.

The statistical significance of the difference in the stiffness was evaluated as shown in Table 10. In terms of stiffness, increasing CRM contents revealed a significant difference in all binders, whereas using the TPU in CRM binders shows a non-significant difference. However, a non-significant difference was witnessed between CRM 5% and CRM 0% + TPU 5%, which means that even though the CRM is applied less with 5%, there is no statistical difference between them due to the effect on decreasing stiffness by using the TPU.

## 4. Summary and Conclusions

To examine the properties of CRM asphalt binders containing TPU, the original binder was artificially aged using short-term and long-term aging. The tests were conducted using the rotational viscosity, the dynamic shear rheometer and the bending beam rheometer to evaluate the properties of binders. Based on the results, the following conclusions were made regarding the effect of TPU on CRM binders in this study.

The addition of CRM raised the viscosity to 135 °C as expected. The CRM binder containing TPU appeared to have higher viscosity compared to general CRM binders. The addition of both additives increased binder viscosity. However, all viscosity results indicated lesser values than the maximum standard suggested by Superpave.

For the DSR test, it is observed that increasing CRM content made it possible to increase G*/sin *δ* in the original condition. Moreover, the addition of TPU into CRM binders raised G*/sin *δ* values. Based on the results, the application of TPU into the CRM binder is beneficial in terms of improving the rutting resistance at its original state.

In the case of RTFO G*/sin *δ*, unlike the trend for original G*/sin *δ*, the CRM binders including the TPU showed the tendency to have less rutting resistance in comparison with their corresponding control CRM binders. This is considered to be due to the TPU contributing to having less of a hard portion in the asphalt binder due to the aging effect after short-term aging.

The binders for the original state were observed to decrease J_nr_ and increase %rec with the addition of CRM. In addition, using the TPU in the original condition made it possible to increase the viscoelasticity of CRM binders, which indicates higher rutting resistance.

In the case of RTFO binders for the MSCR test, all J_nr_ and %rec values showed a similar trend to the data for the original state, with increasing CRM content. However, the viscoelasticity for CRM asphalt binders was observed to be decreased with the addition of the TPU, which is considered to have less of an aging effect to resist the permanent deformation. 

G*sin *δ* value for CRM binders was observed to have a decreasing trend with increasing CRM content. Moreover, addition of TPU into the CRM binder contributed to a relatively lower G*sin *δ* than their corresponding control CRM binders. Therefore, the TPU was found to be effective in improving fatigue cracking performance for CRM asphalt binders. 

The results of the BBR test showed that incorporating a CRM modifier made it possible to decrease the stiffness steadily. The stiffness for the CRM binder containing TPU remained relatively lower than general CRM binders. TPU is considered to play a role in increasing the thermal cracking resistance at low temperatures.

The application of TPU into CRM binders generally showed a potential impact on binder performance, including rutting and cracking. However, the effect of TPU on rutting is not consistent before and after aging. It is necessary to conduct a deeper analysis to figure out the property of TPU with the aging process. 

## Figures and Tables

**Figure 1 materials-15-03824-f001:**
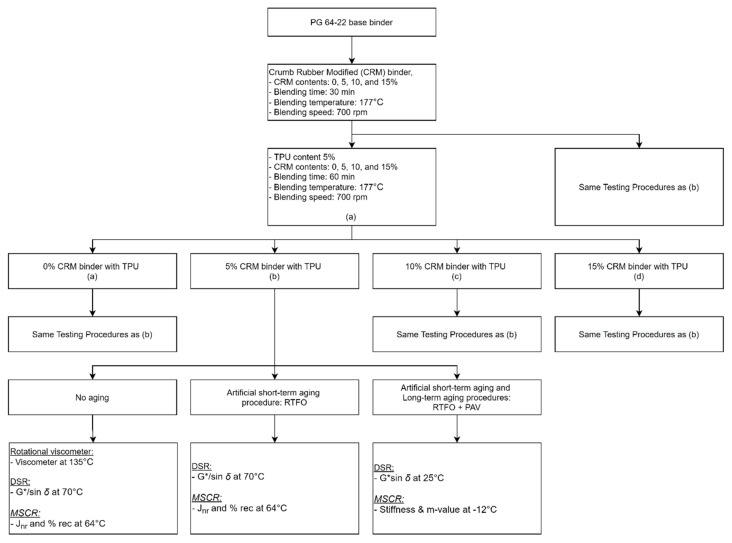
Flow chart of experimental design procedures.

**Figure 2 materials-15-03824-f002:**
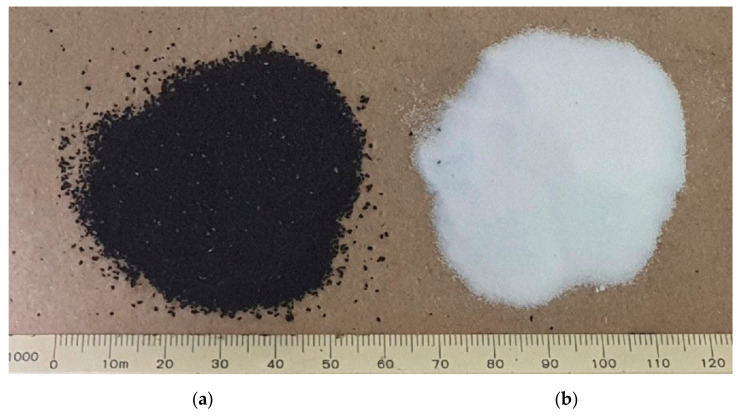
Crumb rubber (**a**) and TPU powder (**b**) used in this study.

**Figure 3 materials-15-03824-f003:**
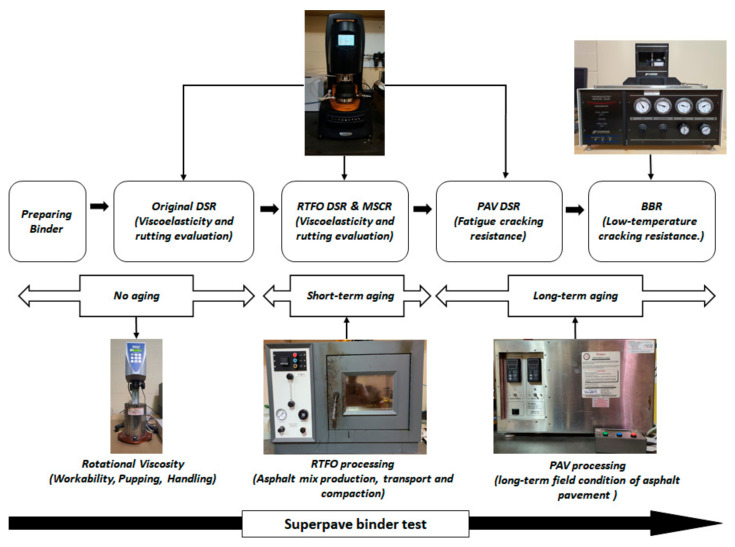
The procedure of Superpave binder test.

**Figure 4 materials-15-03824-f004:**
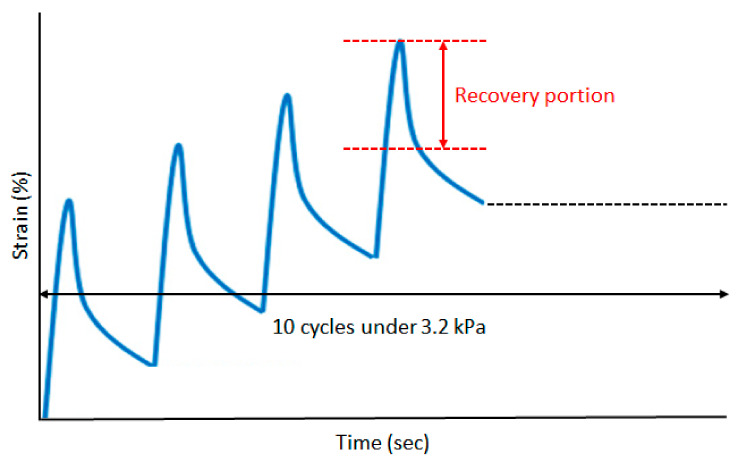
Ten cycles of creep and recovery at 3.2 kPa.

**Figure 5 materials-15-03824-f005:**
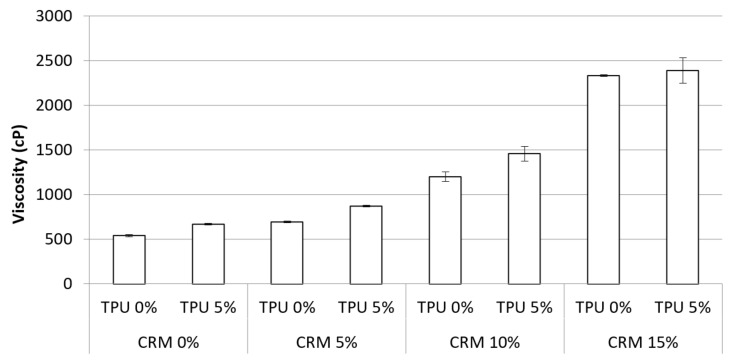
Viscosity of the asphalt binder containing CRM and TPU at 135 °C.

**Figure 6 materials-15-03824-f006:**
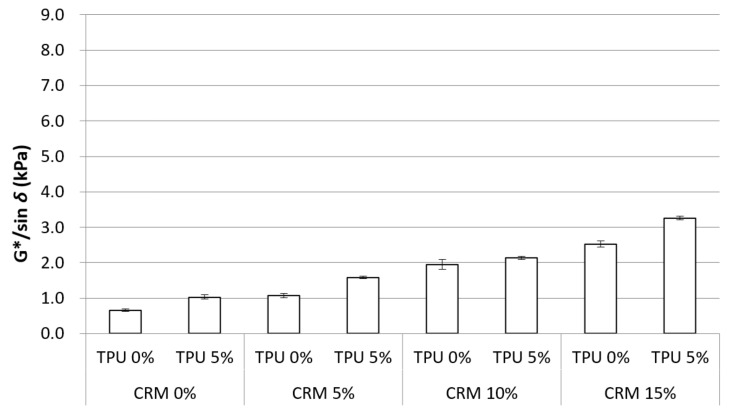
G*/sin *δ* of the asphalt binders containing CRM and TPU for original condition.

**Figure 7 materials-15-03824-f007:**
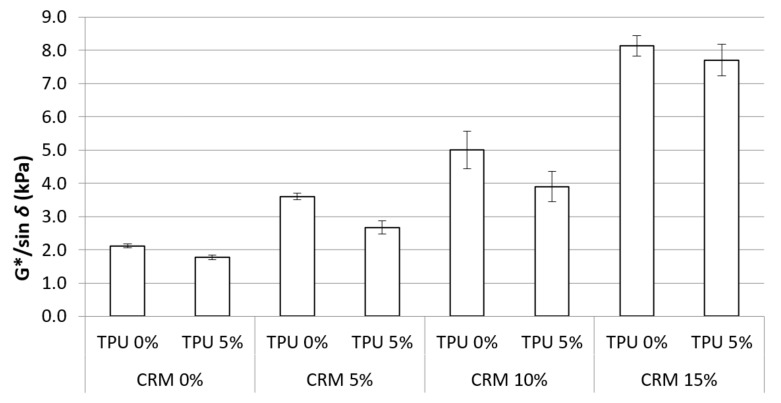
G*/sin *δ* of the asphalt binders containing CRM and TPU for RTFO conditions.

**Figure 8 materials-15-03824-f008:**
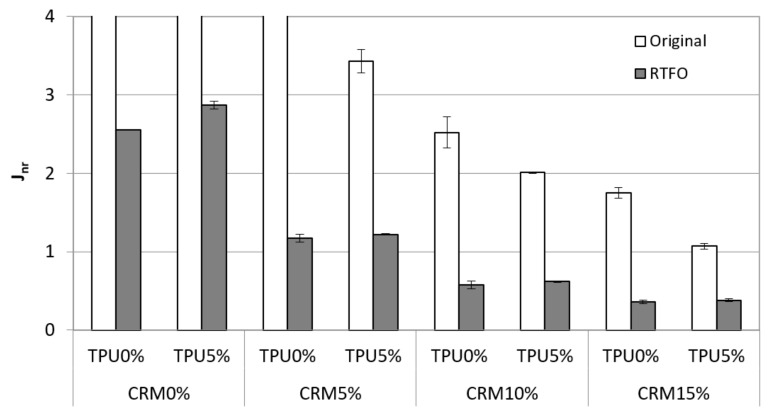
J_nr_ of the CRM asphalt binders containing CRM and TPU for original and RTFO condition.

**Figure 9 materials-15-03824-f009:**
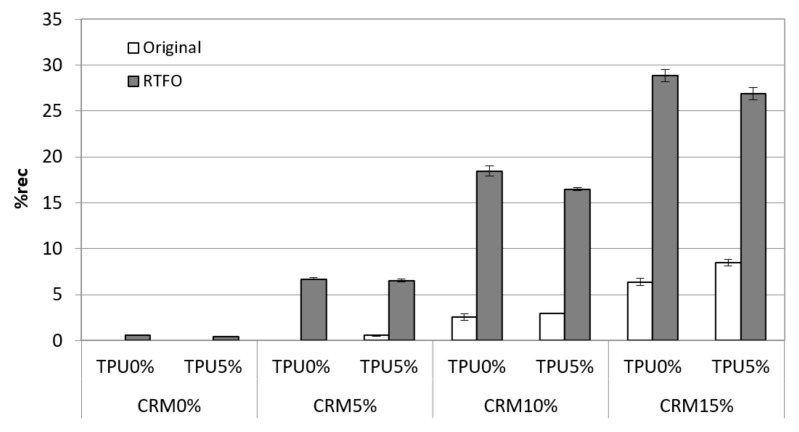
%rec of the CRM asphalt binders containing CRM and TPU for original and RTFO condition.

**Figure 10 materials-15-03824-f010:**
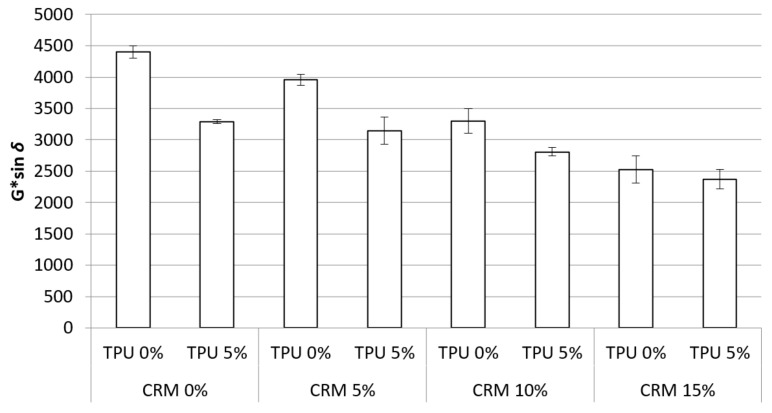
G*·sin *δ* of the asphalt binders containing CRM and TPU for RTFO + PAV condition.

**Figure 11 materials-15-03824-f011:**
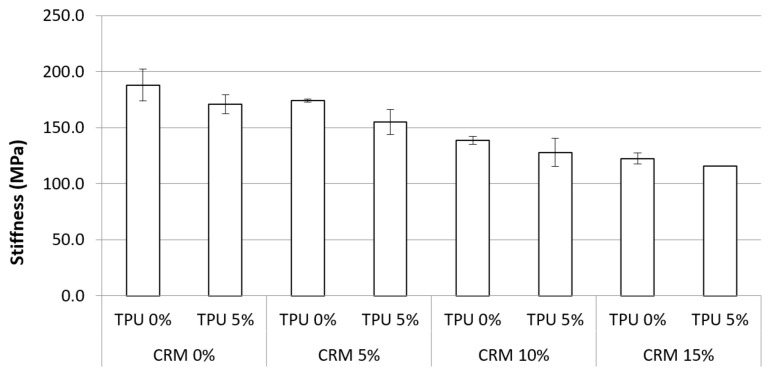
The stiffness of the asphalt binders containing CRM and TPU for RTFO + PAV condition.

**Table 1 materials-15-03824-t001:** Properties of base asphalt binder (PG 64-22).

Aging States	Test Properties	Test Result
Unaged binder	Viscosity @ 135 °C (cP)	538
	G*/sin *δ* @ 64 °C (kPa)	1.38
RTFO aged residual	G*/sin *δ* @ 64 °C (kPa)	3.82
RTFO + PAV aged residual	G*sin *δ* @ 25 °C (kPa)	4402
	Stiffness @ −12 °C (MPa)	205
	m-value @ −12 °C	0.323

**Table 2 materials-15-03824-t002:** Gradation of crumb rubber used in this study.

Sieve Number (μm)	% Cumulative Passed of CRM
30 (600)	100.0
40 (425)	91.0
50 (300)	59.1
80 (180)	26.2
100 (150)	18.3
200 (75)	0.0

**Table 3 materials-15-03824-t003:** Properties of thermoplastic polyurethane (TPU) used in this study.

Properties	Test Method	Units	Typical Value
Density	ASTM D792	lb/in^3^	0.0426
Hardness, Shore A	ASTM D2240	-	67
Tensile strength	ASTM D412	psi	2900
Elongation	ASTM D412	%	700
100% modulus	ASTM D412	psi	435 @ Strain 100%
300% modulus	ASTM D412	psi	725 @ Strain 300%
Tear strength	ASTM D624	pli	371
Abrasion	DIN Abrasion Loss; DIN 53516	-	40
Glass Transition Temp, Tg	DSC	°C	−42.0

**Table 4 materials-15-03824-t004:** Statistical analysis results of the viscosity as a function of CRM contents and the addition of TPU (α = 0.05).

Rotational Viscosity		TPU 0%				TPU 5%			
		CRM 0%	CRM 5%	CRM 10%	CRM 15%	CRM 0%	CRM 5%	CRM 10%	CRM 15%
TPU 0%	CRM 0%	-	N	S	S	S	S	S	S
	CRM 5%		-	N	S	S	S	S	S
	CRM 10%			-	S	S	S	S	S
	CRM 15%				-	S	S	S	N
TPU 5%	CRM 0%						S	S	S
	CRM 5%						-	S	S
	CRM 10%							-	S
	CRM 15%								-

N: non-significant; S: significant.

**Table 5 materials-15-03824-t005:** Statistical analysis results of the original G*/sin *δ* as a function of CRM contents and the addition of TPU (α = 0.05).

G*/Sin *δ* at 70 °C		TPU 0%				TPU 5%			
		CRM 0%	CRM 5%	CRM 10%	CRM 15%	CRM 0%	CRM 5%	CRM 10%	CRM 15%
TPU 0%	CRM 0%	-	S	S	S	S	S	S	S
	CRM 5%		-	S	S	N	S	S	S
	CRM 10%			-	S	S	S	S	S
	CRM 15%				-	S	S	S	S
TPU 5%	CRM 0%						S	S	S
	CRM 5%						-	S	S
	CRM 10%							-	S
	CRM 15%								-

N: non-significant; S: significant.

**Table 6 materials-15-03824-t006:** Statistical analysis results of the RTFO G*/sin *δ* as a function of CRM contents and the addition of TPU (α = 0.05).

G*/Sin *δ* at 70 °C		TPU 0%				TPU 5%			
		CRM 0%	CRM 5%	CRM 10%	CRM 15%	CRM 0%	CRM 5%	CRM 10%	CRM 15%
TPU 0%	CRM 0%	-	S	S	S	N	N	S	S
	CRM 5%		-	S	S	S	S	N	S
	CRM 10%			-	S	S	S	S	S
	CRM 15%				-	S	S	S	N
TPU 5%	CRM 0%						S	S	S
	CRM 5%						-	S	S
	CRM 10%							-	S
	CRM 15%								-

N: non-significant; S: significant.

**Table 7 materials-15-03824-t007:** Statistical analysis results of the J_nr_ as a function of CRM contents and the addition of TPU (α = 0.05).

J_nr_ for Orig.		TPU 0%				TPU 5%			
		CRM 0%	CRM 5%	CRM 10%	CRM 15%	CRM 0%	CRM 5%	CRM 10%	CRM 15%
TPU 0%	CRM 0%	-	N	S	S	N	S	S	S
	CRM 5%		-	S	S	N	S	S	S
	CRM 10%			-	S	S	S	S	S
	CRM 15%				-	S	S	S	S
TPU 5%	CRM 0%						S	S	S
	CRM 5%						-	S	S
	CRM 10%							-	S
	CRM 15%								-
J_nr_ for RTFO.		TPU 0%				TPU 5%			
		CRM 0%	CRM 5%	CRM 10%	CRM 15%	CRM 0%	CRM 5%	CRM 10%	CRM 15%
TPU 0%	CRM 0%	-	S	S	S	S	S	S	S
	CRM 5%		-	S	S	S	N	S	S
	CRM 10%			-	S	S	S	N	S
	CRM 15%				-	S	S	S	N
TPU 5%	CRM 0%						S	S	S
	CRM 5%						-	S	S
	CRM 10%							-	S
	CRM 15%								-

N: non-significant; S: significant.

**Table 8 materials-15-03824-t008:** Statistical analysis results of the %rec as a function of CRM contents and the addition of TPU (α = 0.05).

%rec for Orig.		TPU 0%				TPU 5%			
		CRM 0%	CRM 5%	CRM 10%	CRM 15%	CRM 0%	CRM 5%	CRM 10%	CRM 15%
TPU 0%	CRM 0%	-	N	S	S	N	S	S	S
	CRM 5%		-	S	S	N	S	S	S
	CRM 10%			-	S	S	S	S	S
	CRM 15%				-	S	S	S	S
TPU 5%	CRM 0%						S	S	S
	CRM 5%						-	S	S
	CRM 10%							-	S
	CRM 15%								-
%rec for RTFO.		TPU 0%				TPU 5%			
		CRM 0%	CRM 5%	CRM 10%	CRM 15%	CRM 0%	CRM 5%	CRM 10%	CRM 15%
TPU 0%	CRM 0%	-	S	S	S	N	S	S	S
	CRM 5%		-	S	S	S	N	S	S
	CRM 10%			-	S	S	S	S	S
	CRM 15%				-	S	S	S	S
TPU 5%	CRM 0%						S	S	S
	CRM 5%						-	S	S
	CRM 10%							-	S
	CRM 15%								-

N: non-significant; S: significant.

**Table 9 materials-15-03824-t009:** Statistical analysis results of the G*sin *δ* as a function of CRM contents and the addition of TPU (α = 0.05).

G*Sin *δ*		TPU 0%				TPU 5%				
		CRM 0%	CRM 5%	CRM 10%	CRM 15%	CRM 0%	CRM 5%	CRM 10%	CRM 15%	
TPU 0%	CRM 0%	-	S	S	S	S	S	S	S	
	CRM 5%		-	S	S	S	S	S	S	
	CRM 10%			-	S	N	N	S	S	
	CRM 15%				-	S	S	S	N	
TPU 5%	CRM 0%						N	S	S	
	CRM 5%						-	S	S	
	CRM 10%							-	S	
	CRM 15%								-	

N: non-significant; S: significant.

**Table 10 materials-15-03824-t010:** Statistical analysis results of the stiffness as a function of CRM contents and the addition of TPU (α = 0.05).

Stiffness		TPU 0%				TPU 5%				
		CRM 0%	CRM 5%	CRM 10%	CRM 15%	CRM 0%	CRM 5%	CRM 10%	CRM 15%	
TPU 0%	CRM 0%	-	N	S	S	N	S	S	S	
	CRM 5%		-	S	S	N	N	S	S	
	CRM 10%			-	N	S	N	N	S	
	CRM 15%				-	S	S	N	N	
TPU 5%	CRM 0%						N	S	S	
	CRM 5%						-	S	S	
	CRM 10%							-	N	
	CRM 15%								-	

N: non-significant; S: significant.

## Data Availability

The data used to support the findings of this study are included within the article.

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
