# Peer review of "Evaluation of Effect of Thermoplastic Polyurethane (TPU) on Crumb Rubber Modified (CRM) Asphalt Binder"

_materials, 2022, doi:10.3390/ma15113824_

Round 1

Reviewer 1 Report

General comment: the manuscript studied the rheological performance of of CRM-TPU modified asphalt binder. The subject is not innovative or new but still relevant. The paper needs to be thoroughly edited for grammar and technical terms. Below are some comments for the authors to address.

  1. Kindly provide more detail on how the TPU was blended with the CRM-modified asphalt binder. Because the current mixing conditions (temp. speed, and time) were addressing the addition of CRM only.
  2. Provide the summary of the upper and lower PG of each blend in a tabular form so as to have a clear idea of how each modifier affect the performance of the asphalt.
  3. Authors need to seek professional edit of the paper for grammar and technical terms.

Author Response

Dear reviewer,

Thank you for your comments to develop this research paper. It was really valuable to improve the quality.

Thank you.

Reviewer 2 Report

The experimentation presented in this manuscript is well structured and presented.
The rheological study is complete and follows the consolidated SHRP protocol.

An excellent statistical analysis is conducted to validate the experimental data.

Some suggestions are proposed:

Paragraph 2.2 | The process of adding the CRM and TPU seems to have occurred through the use of a stirrer at 700rpm for 30 min. Generally, this technique of adding solid components produces a high inhomogeneity of the final compound; and this problem occurs with the use of high shear mixers, albeit to a lesser extent.
The authors consult: Elisabeta I. Szerb at Al. (2017): Highly stable surfactant-crumb rubber-modified bitumen: NMR and rheological investigation, Road Materials and Pavement Design, DOI: 10.1080 / 14680629.2017.1289975

How did the authors deal with this problem? It would be necessary to include some comments explaining their work in this section

Figure 1 and Line 104, page 4 | Were the tests using the DSR conducted at 70 ° C or 76 ° C? Align the indication on the manuscript text.

Either 70 ° C or 76 ° C, why was this temperature chosen? The authors explain.

Author Response

(The authors gave the same response as above.)

Round 2

Reviewer 2 Report

The suggestions were considered by the authors.

But beware: all tables and figures are out of format and absolutely not paginated!

Author Response

Dear Author,

Thank you for your comment.

I did my best to revise the format of the tables and figures.

However, the journal initially modified it after I submitted the first version.

I will send the message to the editor regarding this issue related to the format of tables and figures.

Thank you.
